# Traditional Knowledge of Textile Dyeing Plants: A Case Study in the Chin Ethnic Group of Western Myanmar

Tial C. Ling [1], Angkhana Inta [2], Kate E. Armstrong [3,4], Damon P. Little [4,5], Pimonrat Tiansawat [6,7],
Yong-Ping Yang [8], Patcharin Phokasem [1], Za Khai Tuang [9,10], Chainarong Sinpoo [1]
and Terd Disayathanoowat [1,*]

[1]  Bee Protection Laboratory, Department of Biology, Faculty of Science, Chiang Mai University, Chiang Mai 50200, Thailand
[2]  Department of Biology, Faculty of Science, Chiang Mai University, Chiang Mai 50200, Thailand
[3]  Institute of Systematic Botany, New York Botanical Garden, Bronx, New York, NY 10458, USA
[4]  The Graduate Center, City University of New York, New York, NY 10016, USA
[5]  Cullman Program for Molecular Systematics, New York Botanical Garden, New York, NY 10458, USA
[6]  Forest Restoration Research Unit, Department of Biology, Faculty of Science, Chiang Mai University, Chiang Mai 50200, Thailand
[7]  Environmental Science Research Center, Faculty of Science, Chiang Mai University, Chiang Mai 50200, Thailand
[8]  Key Laboratory for Plant Diversity and Biogeography of East Asia, Institute of Tibetan Plateau Research at Kunming, Kunming Institute of Botany, Chinese Academy of Sciences, Kunming 650201, China
[9]  Department of Plant Pathology and Microbiology, The Robert H. Smith Faculty of Agriculture, Food and Environment, The Hebrew University of Jerusalem, Rehovot 76100, Israel
[10]  Department of Botany, University of Kalay, Kalay 03044, Myanmar
*   Correspondence: terd.dis@gmail.com; Tel.: +66-817249624

**Abstract:** Traditional knowledge of the plants used for textile dyeing is disappearing due to the utilization of synthetic dyes. Recently, natural products made from plants have gained global interest. Thus, preserving traditional knowledge of textile dyeing plants is crucial. Here, we documented this knowledge by interviewing 2070 informants from 14 communities of the Chin ethnic group of Myanmar. The Chin communities we interviewed used a total of 32 plant species for textile dyeing from 29 genera in 24 families. *Chromolaena odorata*, *Lithocarpus fenestratus*, and *L. pachyphyllus* were the most important dye species. The most common responses described dyes that were red in color, produced from leaves, derived from tree species, collected from the wild, and used as firewood ash as a mordant to fix the dye to the fabrics. According to the IUCN Red List of threatened species, one species was registered as Data Deficient, 20 species still needed to be categorized, and 11 species were categorized as Least Concern. This study will help re-establish the use of natural dyes, encourage the cultural integrity of the indigenous people, and serve as an example for other communities to preserve their traditional knowledge of plant textile dyes.

**Keywords:** Chin ethnicity; color; ethnobotany; Lutuv-Chin; natural dyes; textile plants

## 1. Introduction

Since prehistoric civilization, plants have been used to maintain humanity's basic life-sustaining needs, including food, fuel, and shelter, and have also been used for furniture, paper, cloth, and natural dyes [1,2]. The first written record of the use of natural dyes extracted from plants was found in China dating back to 2600 BCE [3,4]. Plant parts that have been utilized as dyestuffs include bark, flowers, fruit, leaves, rhizomes, roots, seeds, and the whole plant [4–8]. Other biological sources, such as fungi, lichens, and animals, have also been used [9–11]. Color plays a vital role in the interactions among humans, and the choice of color varies among people, depending on personality traits, color perception, motivation, culture, beliefs, and emotions [12,13]. Therefore, people use different plant

materials from a variety of species to impart the desired colors depending on their interests. Substances, including alum, firewood ash, calcium carbonate, potassium bichromate, and lime, have been used as mordants to bind the dye to the fabric and make it colorfast, as well as to brighten the color [3,14–16]. Throughout history, people have been dyeing textiles using common, locally available materials. Dyes that produce brilliant colors (e.g., purple, blue, green, black, red, and yellow) were rarer and thus considered highly priced luxury items during the medieval period [17], the Ming and Qing dynasties [18], and most Egyptian eras [19,20]. Thus, plant species and the parts used for traditional textile dyeing can vary among communities, depending on their traditional clothes and the locally available plant species.

Despite the importance of natural dyes to traditional cultures, following the invention of Perkin's Mauve in 1856, synthetic dyes became the preferred choice due to their ease of use and efficacy [21,22]. Traditional dyes rapidly became neglected among weavers [22–24] because natural products are more expensive, the method needed to prepare them is more time consuming, and because synthetic dyes are more durable. Such a rapid replacement of natural dyes with synthetics can have a great impact on socioeconomics and productivity. For instance, the use of textiles produced with synthetic dyes has an influence on local people whose income depends on natural dye products. Several studies have suggested that synthetic dyes contain chemical compounds that are harmful to human health [25,26]. It has also had a negative effect on the intergenerational transference of traditional knowledge concerning dye plants and dyeing techniques. Therefore, the use of natural dyes from plant sources has the potential to be a healthier and more sustainable option. The younger generation in most Chin State communities, whose ancestors have relied on natural dyes for centuries, no longer learn these traditional skills (T.C.L.'s personal observation). Specifically, a great decline in such traditional knowledge of natural dyes has occurred among the Chin people in towns because they are able to easily purchase traditional clothes made from either natural or synthetic dyes in the market and do not need to produce the textiles themselves. In contrast, villagers, especially in tribal areas, may still produce textiles utilizing natural dyes extracted from locally available plants. However, empirical evidence is needed to validate these hypotheses. In other words, it is important to record and preserve the knowledge of how natural dyes were traditionally made and what plant sources were used across different communities.

In this study, we documented the traditional knowledge of plants used for textile dyeing in the Chin State of Western Myanmar (Figure 1). The Chin people are one of the founding groups (i.e., Bamar, Chin, Kachin, and Shan) of the Union of Myanmar [27]. The Chin State is culturally diverse and is home to 53 different subtribes and languages of the 135 ethnic groups in Myanmar [27], and they share similar cultures and traditions. In general, local livelihoods are mainly based on paddy rice cultivation in small lowland basins and the shifting cultivation of corn, millet, and vegetables on hillsides. It is postulated that the Chin people migrated to the Chindwin–Irrawaddy plain in Myanmar from China during the first millennium CE and moved westward, where they settled in the present Chin State between the 14th and 16th century [28]. The Chin written language was developed in the late 19th century, with the help of western missionaries. The primary motivation was the translation of Bible scripture and an education in, and conversion to, Christianity. Only in the last half of the 20th century have the Chin begun creating their own written history, and the traditional importance of Chin textiles was formally recorded only in 2005 [28]. Despite this recent effort, the traditional knowledge of dye plants and the textile dyeing process remains poorly understood. A previous study in Thailand documented that only a few Tai-Lao men had traditional knowledge of textile dyeing with plants [5]. In the case of the Chin people in Myanmar, particularly those living in the hills, women practiced a traditional weaving method to produce clothing, while men helped with the collection and preparation of plant materials from the wild for textile dyeing (T.C.L.'s personal observation). Therefore, both genders have the same level of traditional knowledge about the plants that are used as sources of natural dyes, but they participate in different aspects of

dye and textile production. Textile dyeing with locally available plants plays an important role in both the socioeconomic and sociocultural lives of this indigenous ethnic group of Myanmar. The main colors used for traditional Chin clothing are black, green, and red. Several studies have documented that leaves and bark that produce green, red, and reddish brown are the most used plant sources for natural dyes, e.g., refs. [4,5], although the cultural reason for using these plant parts is not known. Traditional dresses are used mainly during important events such as Chin National Day, funerals, religious services at Church, and wedding ceremonies. It is well known that, among Myanmar people, the Chin traditional dress is one of the most favored and expensive textiles due to its intricate weaving design and use of color. In 2021, the beauty pageant contestant for Miss Myanmar, Thuzar Wint Lwin, won the national costume award at the Miss Universe contest wearing a Chin traditional dress, highlighting its national cultural relevance. Although natural dyes continued to be commonly used for cotton and silk fabrics until approximately 1980 (T.C.L.'s personal observation), their use is not likely to continue with the next generation. An additional threat to this traditional plant knowledge is the increasing destruction of plant habitats by shifting cultivation and subsistence farming. These practices have reduced the availability of local textile dye plants. Therefore, an immediate study of the traditional plant knowledge used for textile dyeing by the Chin ethnic group is needed.

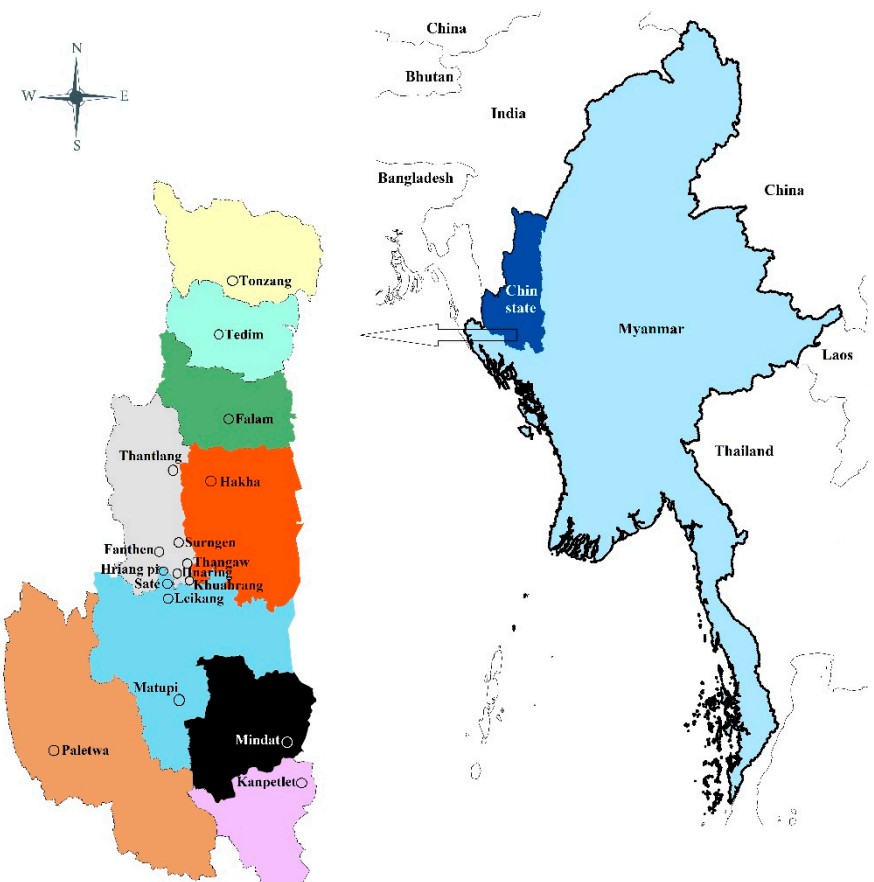

**Figure 1.** Location of Chin State in Myanmar (deep blue color on the left) and the communities, including Falam, Hakha, Kanpetlet, Lutuv (Hnaring, Hriangpi, Khuahrang, Leikang, Sate, Surngen, Thangaw, Matupi, Mindat, Tedim, and Thantlang, where ethnobotanical data related to the traditional textile dyeing plants were collected (left).

In this study, we addressed three specific questions as follows: (I) How many plant species, plant parts, and types of plants were used by local people in the extraction of different dye colors for the production of the Chin traditional dress? (II) What are the most

used plant species and plant parts among the indigenous Chin people for traditional textile dyeing? (III) What is the traditional knowledge of the local plants used for textile dyeing among generations and genders of the Chin people? We hypothesize that various plant parts from several locally available species are used for traditional textile dyeing. Of these, leaves and bark will be the most used plant sources because they produce black, green, and red dyes, which are the colors most used by the Chin people. We also hypothesize that the older generation has more traditional knowledge of textiles dyeing with plants than the younger generation and that there is no significant difference in the knowledge of textile dyeing with plants between genders.

## 2. Materials and Methods

### 2.1. Study Site

The study site was in the Chin State (22°0′31.31″ N, 93°34′52.57″ E), the western part of Myanmar. The state is the most ethnically diverse area in Myanmar [29]. It is bordered by Bangladesh to the southwest, the Indian states of Mizoram to the west, Manipur in the north, Myanmar's Sagaing Region and Magway Region to the east, and Rakhine State to the south (Figure 1). Most of the state is mountainous, and peaks can reach up to 3000 m above sea level (i.e., Nat-Ma-Taung National Park) [30,31]. The landscape forms part of the Chin Hills' warm temperate rainforest ecosystem [32,33], which is composed of a humid warm temperate and evergreen subtropical moist broadleaf forest and is home to many Himalayan species as well as many endemics [31].

Surveys were conducted in 14 communities (8 towns and 6 villages), including Hakha (the capital of the state), Falam, Kanpetlet, Lutuv (i.e., Hnaring, Hriangpi, Khuahrang, Leikang, Sate, Surngen, and Thangaw), Matupi, Mindat, Thantlang, and Tedim (Table S1). Prior to our present study, we noted that most people in town did not know about traditional dye plants and natural textile dyeing methods. Therefore, we extended our study to six villages (Table S1) in the Lutuv community, where traditional knowledge of natural dyes still exists. There are four reasons we chose to study these villages in Hnaring of the Lutuv community, known by other communities as the Laotu-Chin: (I) several people from the older generation (>50) of Lutuv-Chin still practice or know about natural dyeing methods and the local plants used for textile dyeing; (II) people from other towns in the Chin State are more educated than those in the Lutuv community, and this might result in differences in the traditional knowledge among selected communities; (III) the first author and his family are originally from the Hnaring-Lutuv community and have a background in utilizing natural dyes and weaving fabric to make clothes; and (IV) due to travel restrictions in Myanmar during the study period, access to a wider array of Chin communities was not possible—only access to the Lutuv-Chin community area was granted at this time.

### 2.2. Interview and Plant Material Collection

From January 2022 to July 2022, we interviewed 2070 (1083 females and 987 males) local informants (Table S2). We used snowball sampling to recruit the informants [34]. The informants who agreed to participate were mostly weavers or those who had a family background of utilizing plant-based natural dye to fabricate clothes. Selected informants were divided into five categories based on their age as follows: 187 informants aged ≥60, 400 informants aged between 50 and 59, 446 informants aged between 40 and 49, 495 informants aged between 30 and 39, and 542 informants aged <30. The age of the selected informants ranged from 20 to 85 years old. We selected 12 to 46 local informants in each generation per community. We recorded the education level, such as illiterate, primary, high school, and university, of each informant. Free lists from each informant who knew about natural dyes were elicited by the prompt: "Which plant sources are used for dying fabrics? What are the specific uses of this plant in the dyeing process?" to obtain frequency. We recorded the number of known textile dyeing plants, Burmese name, collection season, color production, dye uses, local name, plant life form, mode of use, parts of the plants used, and the harvest season of each plant species. Next, we recorded the extraction,

preparation, and dyeing process of plant dyes through participatory investigation and field investigation, followed by the collection of voucher samples and photographs of each voucher sample. We collected plants that were used as dye sources from the study area or nearby localities with the help of local informants. Permission was granted by villagers to collect specimens on their land, and all specimens were kept in Myanmar at Kalay University herbarium. As this was a local study and was not based within a protected area, collecting permits were not required. We then made herbarium specimens from the voucher samples and identified each specimen at the species level (i.e., botanical name) with the help of plant taxonomists. Plant specimens were deposited at the Herbarium of Kalay University, Kalay, Myanmar.

*2.3. Quantitative Ethnobotanical Indexes*

The use categories (i.e., Use Report and Use Value) used in this study are mordant, black, blue, brown, green, purple, red, and yellow.

2.3.1. Use Report (UR)

The URs were counted for each plant species as follows:

$$UR_s = \sum_{u=u_1}^{u_{CN}} \sum_{i=i_1}^{i_N} UR_{ui}$$

The *URs* calculated the total uses for the species by all informants from $i_1$ to $i_N$ within each use category for that species (s). It is a count of the number of informants who mention each use-category NC for the species and the sum of all uses in each use category from $u_1$ to $u_{NC}$ [35,36].

2.3.2. Use Value (UV) Index

The UV was used to indicate species that are considered highly important using the following formula:

$$UV_s = \sum_{i=i_1}^{i_N} \sum_{u=u_1}^{u_{NC}} UR_{ui/N}$$

*Ui* is the number of different uses mentioned by each informant, and *i* and *N* is the total number of informants interviewed in the survey. The UV index varied from 0, when nobody mentioned any use of the plant, to 1, when the plant was most frequently mentioned as important in the maximum number of use categories [35,36].

*2.4. Data Analyses*

All statistical analyses were conducted using R version 4.1.2 (R development Core Team, 2022). All indices used for quantitative ethnobotanical analyses were performed using the ethnobotany R package (R codes for indices and data were provided in 3S and S4) [36]. We excluded reports by informants with the age of <40 years old from the towns and villages of Lutuv-communities because the data were insufficient to run meaningful statistical analyses. The traditional knowledge of the plants used for textile dyeing and the number of local plant species known by the informants (age between $\geq$60 and $\geq$40) were compared among ages and between genders using a generalized linear model (GLM), with Binomial errors for the proportion data and Normal errors for the count data for the response variables. Prior to the analyses, we calculated the mean proportion of traditional knowledge of textile dyeing reported by individual informants for each community. Age and gender were used as explanatory variables, and their interactions were also included. The data for the informants < the age of 40 and living in cities were insufficient to run statistical analyses. We thus analyzed only the data for the informants $\geq$ the age of 40 and the Lutuv-communities. Data are expressed as the mean $\pm$ standard errors.

## 3. Results

*3.1. Documented Plant Species and Mode of Uses and Preparation*

A total of 30 species from 29 genera and 24 families were recorded among the informants (Appendix **??**). We were not able to collect specimens of two species, which are used as mordants, and therefore we kept them as unknown species. Only one or two informants from the towns, including Falam, Hakha, Kanpetlet, Matupi, Mindat, Tedim, and Thantlang, knew about traditional dyes, but they were not able to mention plant species and how they were used. Most information about natural dyes was received from the older generation (>50) of the Lutuv-Chin people, such as Hnaring, Hriangpi, Khuahrang, Leikang, Sate, Surngen, and Thangaw. The coordinates and map of study sites are shown in Supplementary Table S1 and Figure 1.

The most frequently extracted color among the Lutuv-Chin was red (41.0%: Figure 2C,D), followed by black (19.1%), brown (17.6%), green (12.9%: Figure 3C,D), yellow (5.8%), purple (2.9%), and blue (0.73%) (Figure 4A). Local people used various parts of the plants for color extraction, in which the leaves (28.5%: Figure 4B) were the most utilized plant part, followed by bark (25.7%) (Figure 4B), fruit (18.1% each), the whole plant (13.6% each), rhizome (7.0%), seed (4.2%), and flower (2.9%) (Figure 4B). The utilization of plant life forms showed that the most documented species were trees (45.4%) followed by herbs (24.6%), shrubs (22.8%), and climbers (7.2%) (Figure 4C). In total, 58.6% of textile dyeing plants were collected from the wild, whereas 41.4% were produced from cultivated species (Figure 4D). The materials for extracting the color were prepared by crushing (Figure 3B), boiling, soaking, and pulverizing (see the mode of uses in Appendix **??**).

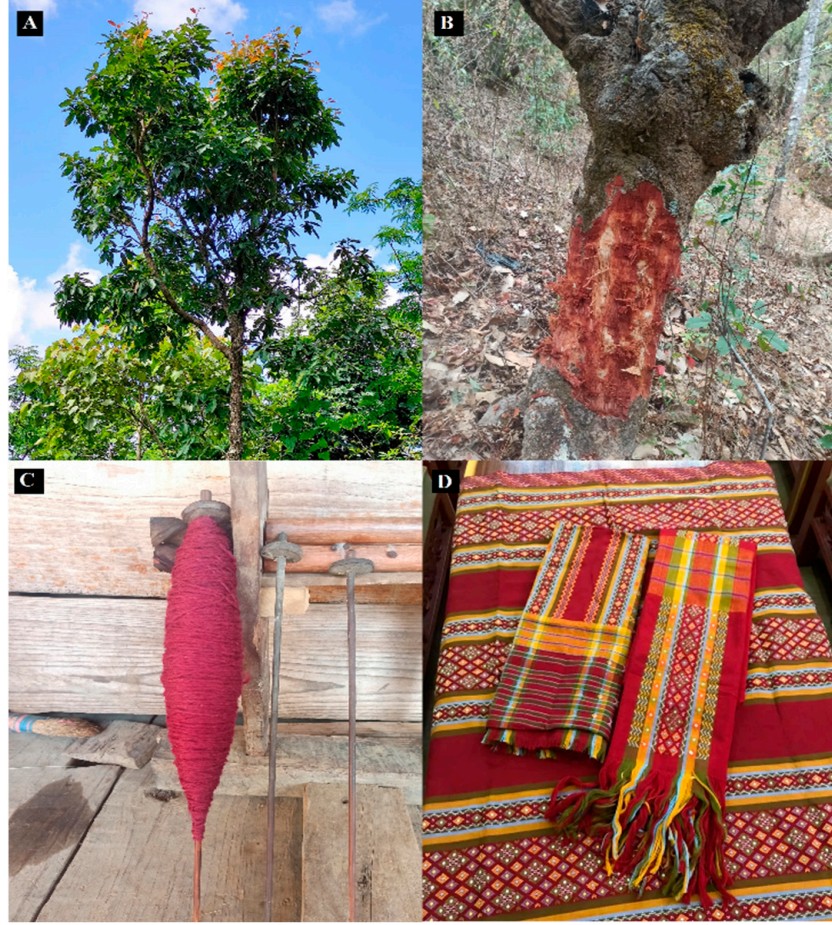

**Figure 2.** (**A**) *Lithocarpus fenestratus* tree and (**B**) trunk with a red inner bark. (**C**) Red dye on cotton obtained from *L. fenestratus* bark and (**D**) Lutuv-Chin fabric prepared with plant dyes.

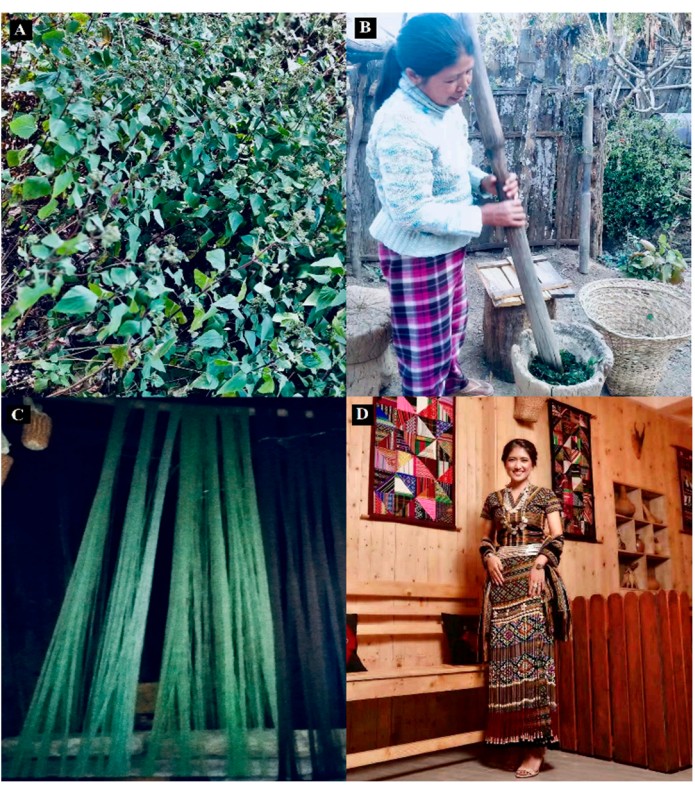

**Figure 3.** (**A**) *Ageratina adenophora* plants, (**B**) crushing the leaves of *A. adenophora* using a traditional mortar and pestle (also known as sung-cuo and sung-khe in Lutuv: Chin) before boiling with water by Ms. Cherry in Hnaring, (**C**) cotton thread dyed green with *A. adenophora* leaves, and (**D**) Ms. Deborah Van Dawt Sung wearing traditional dress.

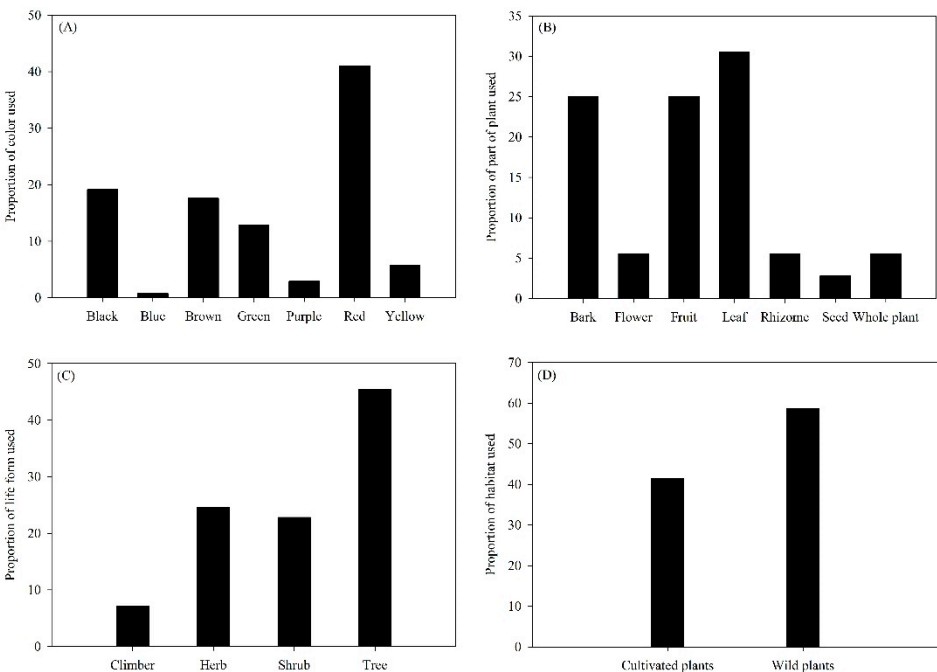

**Figure 4.** Number of (**A**) colors, (**B**) plant parts, (**C**) life forms, and (**D**) habitats used as sources for textile dyeing by the Lutuv-Chin community.

### 3.2. Quantitative Ethnobotanical Analyses

The Use Report (UR) ranged between 590 and 1 and the Use Value (UV) index ranged between 2 and 0.00. Of these, the highest UR and UV values were *Chromolaena odorata* (L.) R.M. King and H. Rob., followed by *L. pachyphyllus* (Kurz) Rehder (Figure 2A,B), *Lithocarpus fenestratus* (Roxb.) Rehder, *L. dealbatus* (Hook.f and Thomson ex Miq.) Rehder, and *Bixa orellana* L., whereas the lowest UR and UV values were found for *Myrica esculenta* Buch. -Ham. ex D. Don (Appendix **??**).

### 3.3. Traditional Knowledge of Plants Used in Textile Dyeing by Age and Gender

Only 309 out of 2070 informants, including all towns and villages across different ages, had traditional knowledge of the textile dyeing method. There was a negative correlation between the traditional knowledge of the plants used for textile dyeing and the ages of the informants (Wald = 126.09; df = 2; $p < 0.001$). The younger generation had less knowledge of textile dyeing plants than the older generation ($0.74 \pm 0.03$ for age $\geq 60$, $0.49 \pm 0.03$ for the age of 50 to 59, and $0.30 \pm 0.03$ for 40 to 49; Figure 5A). There was a significant difference in the traditional knowledge of textile dyeing with plants between genders ($0.64 \pm 0.02$ for female; $0.37 \pm 0.02$ for male; Wald = 66.437; df = 1; $p < 0.001$), but the interaction was not significant (age $\times$ gender: Wald = 0.164; df = 2, $p = 0.921$).

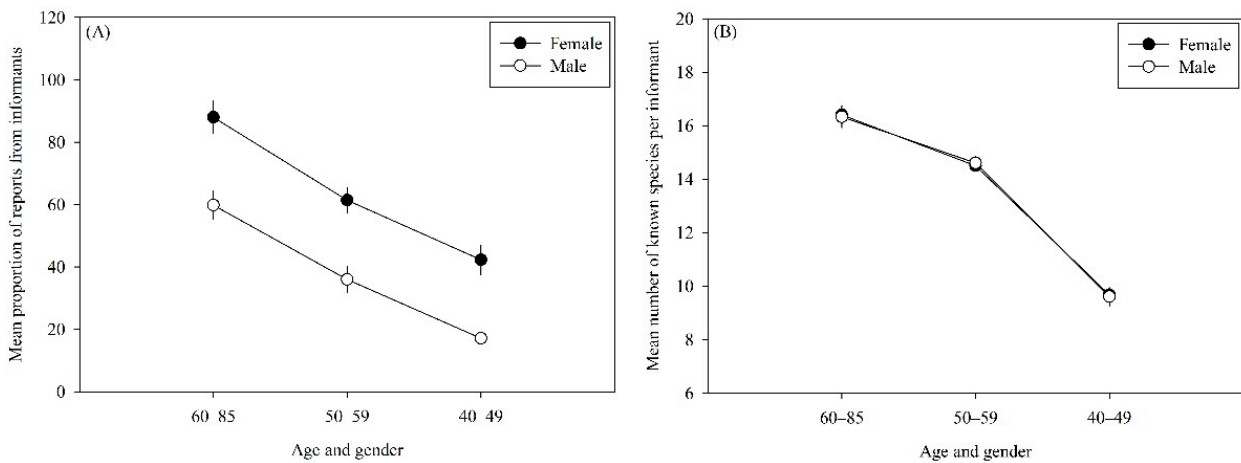

**Figure 5.** (**A**) Proportion of informants who had knowledge of textile dyeing with local plants and (**B**) mean number of local textile dyeing plants known by each informant. 60 to 85, 50 to 59, and 40 to 49 represent each categorized generation age. Bars represent mean $\pm$ SE.

The number of textile dye methods and plant species known by selected informants varied by age (Wald = 518.82; df = 2; $p < 0.001$). The older generation knew more about local plant species than the younger generation did ($16.37 \pm 0.21$ for age $\geq 60$, $14.56 \pm 0.17$ for the age of 50 to 59, and $9.64 \pm 0.22$ for the age of 40 to 49, Figure 5B). When we asked the informants who knew textile dyeing methods and plants, there was no significant difference in the number of known textile dye plants between genders ($13.53 \pm 0.14$ for female; $13.52 \pm 0.19$ for male; Wald = 0.003; df = 1, $p = 0.958$) and the interactions were not significant either (age $\times$ gender: Wald = 0.155; df = 2, $p = 0.925$).

## 4. Discussion

### 4.1. Lutuv-Chin Textile Dyes

The Lutuv-Chin, an indigenous people inhabiting the southern Chin State of Myanmar, share the culture and lifestyle of other Chin communities [27]. In the Chin State, organic cotton and silk are the primary textiles used for natural dyeing, and cloth is woven from these dyed textiles. During the period when natural dyeing was still practiced by the Chin, fabric and clothes were generally made by individual households for family use only, not as commercial products. Traditionally, the dyeing and weaving processes were thought

to be a woman's job. In this study, the number of female informants was slightly higher than that of male informants. Counter to our hypothesis, the proportion of reports from informants who knew textile dyeing methods was significantly higher in female informants than in male informants. Nevertheless, this result is consistent with the study on Lao-Thai textile dyes in northern Thailand, where traditional knowledge of the plants used for textile dyeing was held mainly by female informants [5]. However, when we asked the informants, male informants knew as many dye plant species as female informants. The two contrasting results suggest that some men belonged to a household did not participate in collecting plant materials or traditional dyeing, whereas those having knowledge of the dyeing process might have collected plant materials and thereby knew as many local plant species as the women did. These findings could also be biased as a result of the informants that we selected in this study. During the study period, only five 45 to 50 years old informants from Hnaring were found to be still making natural dyes extracted from *Chromolaena odorata* (L.) R.M. King and H. Rob., *Lithocarpus fenestratus* (Roxb.) Rehder, *L. dealbatus* (Hook.f and Thomson ex Miq.) Rehder, and *Curcuma longa* L. plants. Their motivation for making these natural dyes was due to a request from a Chin cotton textile company. Therefore, it seems likely that natural textile dyeing is no longer practiced by the Chin people of Myanmar unless there is a market demand due to cultural and ecotourism. Unfortunately, most informants under the age of 40 did not have the traditional knowledge of dye plants and the dyeing process, suggesting that this knowledge has been gradually disappearing over the last three to four decades. It seems likely that the older generation did not pass this invaluable knowledge to the younger generation. On the other hand, such a rapid decline in traditional knowledge of the plants used for textile dyeing among the young Chin generation could be a consequence of the replacement of natural dyes with synthetics, which are cheaper and easy to access. Based on our surveys, the young generation was not interested in either traditional weaving or natural dyes, which ultimately affects their traditional knowledge, cultural integrity, and livelihood.

### 4.2. The Lutuv-Chin Dye Plants

In the present study, we found that the Fabaceae and Fagaceae families had the largest number of species used for textile dyeing in the Lutuv-Chin community of Myanmar. Previous studies on textile dyes in Tripura [37], Manipur [7,38], and Mizoram [4] in Northeast India, which borders Chin State, also documented that Fabaceae and Fagaceae were the main sources of plants used for textile dyeing. Moreover, the Tai-Lao community in Thailand used species from the family Fabaceae as the main source for textile dyeing [5]. However, our results are incongruent with reports from Malaysia where Euphorbiaceae and Arecaceae represented the largest number of species used as dye plants [39,40]. The similarity in the most used plant family for textile dyeing among the indigenous people of Chin-Myanmar, Mizo-India [4] and Manipuri-India [38], and Tai-Lao [5] may be due to the high similarity of the anthropogenic landscapes, customary approach, sociocultural context, and vegetation types. However, further studies are needed to validate these hypotheses. In contrast, the dissimilarity in the species used for the textile dyes between the study areas in Myanmar and Malaysia [39] could be due to differences in plant source availability. For example, the Lutuv-Chin forests in our study site were dominated by subtropical to temperate species where Fabaceae and Fagaceae are very common elements of the vegetation, whereas the study area in Malaysia had tropical rainforest habitats where both Euphorbiaceae and Arecaceae are dominant [40]. There have also been several studies that used a wide variety of plant species without emphasis on a specific plant family [6–8]. Moreover, two studies conducted in the same regions of Manipur state in India reported different plant species from different plant families, such that Kikim et al. [26] found Fagaceae as the main plant family used for textile dyeing, whereas Potsangbam et al. [7] did not cite a specific plant family as being the most utilized. The cause of the different species and families reported by these two studies is not clearly known. However, it is possible that this pattern could

result if the collection time and the informants, or the style and the type of interview (e.g., locals versus outsiders) were different between the two studies.

*4.3. Color Production from Lutuv-Chin Dye Plants*

The Lutuv-Chin used their own custom approach to extract various colors from plant sources and dye textiles. In general, the plant materials used for textile dyeing are ground or crushed using a mortar and pestle, and then boiled in water with the mordant to obtain the desired fabric color (detailed information for the mode of uses is provided in Appendix **??**). As has been found by many other studies [4,8,29], they prepare the plant materials used for extracting color by crushing and boiling with water. This approach is similar to the methods used in Mizoram and Manipur of India and many Southeast Asian countries [10,14,38,40,41]. The Lutuv-Chin extracted six main colors from seven different plant parts. In Mizoram, six different colors were extracted from six different plant parts [4], and many of the plant sources they used were similar to the Lutuv-Chin. It has been documented that the Mizo, Manipuri, and Chin share tribal traditions with a long history [28–42]. This gives weight to the hypothesis that these indigenous people may have used the same plants and traditional methods for textile dyeing when their land was contiguous before they migrated up the mountainsides. It seems likely that the textile dyeing tradition has persisted among these hill tribes and remained a crucial element in the expression of their cultural identities through customary clothing.

In many countries, the same species and parts of plants are used to produce more than one color [3,41,43,44]. For example, the people of Roi Et province in Thailand produced a brown–golden–yellow color from *Azadirachta indica* A. Juss. bark, whereas this species in Ghana produced a red–brown color [43] and fairly dark shades in Uganda [44]. In Northeast India, the timing of the decoction was adjusted to obtain the desired shade of color [4,7,38]. Consistent with this trend, the Lutuv-Chin communities often mixed up to three different plant species to obtain the desired colors or to produce more than one color from a single plant species by adjusting the timing of extraction from plant sources. For example, a reddish brown is obtained by boiling the bark of *Albizia chinensis* (Osbeck) Merr. For one to two hours, but a darkish brown color is obtained when the bark of this species is soaked in water for several days. It has also been suggested that different colors can be produced from the same plant using different mordants and methods. The large differences in these colors could be related to changes in the chemistry of local plants, which might be influenced by local environmental factors and microhabitat soils as well as by the types of mordants and the length of time the cloth is left in the dye bath. Further studies may focus on such possibilities.

Clothes made from plant-based natural dyes have declined globally. Recently, only few people (two out of 2070 local informants) have begun to produce new colors due to market demand from cultural and ecotourism [5]. This potential source of income might have encouraged the local weavers to search for new colors to dye their cotton and silk products. However, the important roles of traditional plant dyes and dyeing methodologies are still likely to be neglected by local people, and synthetic dyes are likely to be used instead.

Leaves were the most used plant part for dye extraction for Lutuv-Chin fabrics and clothes. The predominant use of leaves has also been reported in Mizoram, India [4]. In contrast, the bark was the primary plant part used for dye extraction in Manipur, India [38], and Southwestern China [8], whereas trees were reported to be the most common life form for textile dyes in other studies [5,6,8].

The dye plant species with the highest values obtained from the Use Report and the Use Value indices were *Chromolaena odorata*, *Lithocarpus fenestratus*, and *L. pachyphyllus*. These indicate that the three species are the most important and favored within the Lutuv-Chin community. *Albizia chinensis*, *Bixa orellana* L., *Curcuma longa*, *L. dealbatus*, and *Morus nigra* L. were still important and well-utilized plant species within this community. *Lithocarpus* species are the dominant trees in the study sites and are largely used by the local people for firewood, shelter, and textile dyeing. The *Lithocarpus* species, *Albizia chinensis*, *Morus*

*nigra*, and *B. orellana* (seeds) were used to extract red to reddish-brown dye. The leaves of *Chromoleana odorata* and *Bixa orellana* were used to extract green and black colors, whereas *Curcuma longa* was used to extract a yellow color. These above-mentioned colors and plants are the most used sources for Chin textiles. In contrast, the textile dyeing plant species with the lowest Use Value in the relative importance index were *Myrica esculenta Buch*. Ham. ex D. Don, *Mentha spicata* L., *Solanum violaceum* Ortega, and *Parkia lieophylla* Kurz, and the results were consistent across all the calculated indices. These four species produced reddish brown, green, purple, and dark brown colors. Plants with low Use Values are not necessarily unimportant, but this result may reflect that traditional knowledge about them is at risk of not being transmitted and that it may be gradually disappearing, or the low Use Value of some plant species could be related to their local scarcity [5,45,46].

A number of the most used plant species for textiles are non-native to Myanmar. For example, *Ageratina adenophora* (Spreng.) R.M. King and H. Rob. [47], *Bixa Orellana* [48], and *Chromolaena odorata* are from South America [49]; *Curcuma longa* is from India [50]; *Hibiscus sabdariffa* is from Central Africa [51]; *Mentha spicata* is from Europe [52]; and *Morus nigra* is from Iran [53]. In addition, some of these species were introduced to Southeast Asia in the mid-18th century, such that *Chromolaena odorata* was introduced to tropical Asia in the mid-1800s [49] and *Ageratina adenophora* was introduced to Yunnan, China around 1940 [54]. These species were introduced to Asia after the Chin people settled in the present Chin State in Myanmar, which was believed to be between the 14th and 16th centuries [28]. Moreover, before their arrival in the Chin hills, these people were thought to have settled in lowland areas of the Chindwin valley during the first millennium CE [28]. These two regions have very different vegetation types, where the Chin hills support temperate and subtropical deciduous forests and the Chindwin valley has a subtropical evergreen forest. In general, the ecology and plant species between these regions are different due to the change in elevation between the regions. The historic patterns of plant use and color preference by the Chin ethnic group may differ from century to century, depending upon the locally available plant species. For example, the Chin might have used indigenous species that produced a similar color to those of invasive species before their arrival in the Chin hills, although it may never be known. It is also important to note that the present results were obtained only from informants in the Lutuv-Chin communities in Myanmar, where red and dark colors are mainly used for traditional dresses. In fact, traditional clothes and the use of color vary among Chin communities, where green or purple textiles are favored by the Mindat-Chin people and red and dark colors are favored by the Hakha-Chin. Therefore, we cannot confirm whether red is the most used color without studying historical patterns and the use of color within the entire Chin community across different centuries.

The use of textile dyes extracted from plant sources persisted in the Lutuv-Chin communities until 1980 (T.C.L.'s personal observation) and it is an important element in the expression of the Lutuv-Chin communities' cultural identity through customary clothing. Some of the 30 identified textile dyeing species were also used for medicinal purposes. For example, the leaves or bark of *Parkia leiophylla* and the powder of *Curcuma longa* are used to cure stomach pain, while the fruit or leaves of *Hibiscus sabdariffa* L. are used as a cooked vegetable and to make a beverage, respectively.

The Lutuv-Chin used more wild plants than home garden plants as textile dyes. This is in contrast with the Tai-Lao community in Thailand [5], where plants used for textile dyeing were collected from home gardens, or at least the local people collected wild plants and cultivated them for greater accessibility. In the present study, just one of the thirty identified plant species used by the Lutuv-Chin as a textile dye was registered in the IUCN Red List of threatened species as Data Deficient, while eighteen species were not yet categorized (Appendix **??**). In addition to these Red List species, *Strobilanthes cusia* (Nees) Kuntze is a locally rare species, which is used as a source of black dye—one of the most favored colors of the Chin people. In general, the Chin people in Myanmar have long utilized shifting cultivation for food security. Although most dyeing plants are not reported to be under any threat at present, such deforestation and exploitation of raw materials from the forests of

the Chin hills has had a great impact on the loss of sources of plant textile dyes. Therefore, managing these resources could help the Chin people conserve natural dye plants and the habitats in which they grow. Their sustainable use could thus assist in the promotion of biodiversity conservation.

*4.4. Lutuv-Chin Use of Mordants*

Using a mordant or dye fixative while processing textiles for the dyebath is a common practice in several countries [3,5,14,40]. In our study, ash derived from the *Lithocarpus* species, as well as the liquid derived from the ground leaves of an unknown species with water (Appendix **??**) were used as mordants in the Lutuv-Chin community. Some local informants reported that the use of ash as a mordant depends on the types of plants used for the firewood. In general, the Lutuv use the *Lithocarpus* species as firewood not only because they are common locally but also because shifting cultivation is practiced by this community, where trees are cut down and used as firewood before cultivating crops (e.g., corn, rice, cucumbers, and pumpkins). Since the late 20th century, the Chin people, particularly those living in the tribal areas of the Chin hills, have mostly eaten corn for food, and it is often referred to as their traditional food. Several hours are required to completely cook corn food, meaning that a large amount of firewood is used in the process. In turn, the firewood produces an adequate amount of ash for use in dyeing. Some informants also believe that when used as a mordant, the ash of these species provide a better depth of color, wash fastness, and light fastness. By using different mordants, a variety of shades and different colors can even be obtained from a single dye. Many communities in other countries also use wood ash or other chemical substances as mordants during the dyeing process.

*4.5. Limitations of Study and Recommendations*

Understanding traditional textile dyeing and color extraction from local plants is crucial to the Chin people because their cultural identity is expressed through traditional clothing. As mentioned earlier, except for the Lutuv-Chin community, most selected informants from the towns, particularly the young generation, had no traditional textile fabrication knowledge. It is not clear whether the results of our current study entirely represent the total number of plants used for traditional dyes. Therefore, it is worthwhile to continue studying the traditional knowledge of natural dyes with other communities in additional locations. An urgent study is needed to validate this information, as the present research showed that only the older generation over the age of 50 had in-depth knowledge of traditional textile dyeing techniques. In addition, further study is required to improve the extraction technique and the stability of the dye to evaluate the phytochemical properties, bioactivity, and safety of pigments and to orient these resources and their associated culture towards the development of sustainable livelihoods, as has been performed in China [55]. In so doing, a pilot unit can be developed to produce the pigments on a larger scale to sustainably supply users nationally and internationally.

A rapid change in culture and socioeconomics around the globe has had a great impact on the indigenous weavers and natural dyers in the Chin State, resulting in a reliance on synthetic dyes and neglect of natural dyeing methods. This could induce a rapid loss in ethnic textile knowledge in this state. In addition, shifting cultivation is still widely practiced and firewood is used in many rural places throughout the state. Such resource exploitation could lead to the extinction of rare species. Therefore, proper management and forest conservation initiatives are also needed to preserve traditional ethnic knowledge and to sustain their economic and cultural heritage.

**Supplementary Materials:** The following supporting information can be downloaded at https://www.mdpi.com/article/10.3390/d14121065/s1: Table S1: Locations and names of communities of collected natural dye based on plants in Chin State of Myanmar; Table S2: Demographic characteristics of informants from nine communities. The R-script and data for the use categories used in this study are mordant, black, blue, brown, green, purple, red, and yellow, and they are available in the text document (Rcode for indices_S3) and excel file (combined_data_S4). Appendix S1. A total of 30 species from 29 genera and 24 families were recorded among the informants

**Author Contributions:** Conceptualization, T.C.L. and T.D.; methodology, T.C.L., A.I., K.E.A., D.P.L., Z.K.T. and Y.-P.Y.; software, T.C.L.; formal analysis, T.C.L.; investigation, T.C.L. and T.D.; resources, T.C.L. and T.D.; data curation, T.C.L.; writing—original draft preparation, T.C.L., A.I., K.E.A., D.P.L., Z.K.T. and Y.-P.Y.; writing—review and editing, T.C.L., A.I., K.E.A., D.P.L., P.T., Y.-P.Y., Z.K.T., P.P., C.S. and T.D.; visualization, T.C.L.; supervision, T.D.; funding acquisition, T.C.L. and T.D. All authors have read and agreed to the published version of the manuscript.

**Funding:** This work was supported by the Mekong–Lancang Cooperation Special Fund 2022.

**Institutional Review Board Statement:** Not applicable.

**Informed Consent Statement:** Informed consent was obtained from all subjects involved in the study.

**Data Availability Statement:** The data that support the findings of this study are available from the corresponding authors upon request.

**Acknowledgments:** The authors thank three anonymous reviewers for their constructive comments on an earlier draft of this manuscript. We also thank Deborah Van Dawt Sung (Surngen, Chin State, Myanmar) for providing her beautiful picture, Cherry and Thla Cin (Hnaring, Chin State, Myanmar) for the helpful field surveys and for providing important information. T.C.L. particularly thanks his family Tlang Tial (mother), Tin Ung (father), Daisy Van Zing (sister), Tial Lian Thang (brother), Hnin Thandar Aung (late sister), Lin Maung (cousin brother), and Thang Pum (uncle) for their invaluable help in collecting important data, field surveys, and plant specimen collection. We particularly thank Sui Ze (T.C.L.'s grandmother) for providing invaluable information on the traditional dyes in Hnaring and all informants for providing their time and useful information. We also thank the local authorities for allowing us to conduct this important research. T.C.L. also thanks Nina Regina M. Quibod (XTBG, CAS, China) for helping with the geographic distribution range map using ArcGIS. T.C.L. is supported by the postdoctoral fellowship 2022, office of research administration, Chiang Mai University, Thailand, grant number R000030568.

**Conflicts of Interest:** The authors declare no conflict of interest. The funders had no role in the design of the study; in the collection, analyses, or interpretation of data; in the writing of the manuscript; or in the decision to publish the results.

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
