# Peer review of "Traditional Knowledge of Textile Dyeing Plants: A Case Study in the Chin Ethnic Group of Western Myanmar"

_diversity, doi:10.3390/d14121065_

Round 1

Reviewer 1 Report

This is an interesting paper, investigating the traditional knowledge of textile dyeing plants by the Chin ethnic group of Myanmar. I really enjoyed reading it and learning from it. I made various comments on the PDF (mostly minor edits or suggestions). However, I feel that the manuscript requires further work to be accepted in this journal.

Introduction

It would be useful to expand on the differences between towns and villages in the local context. This is key to understanding why in towns the traditional knowledge has been lost, compared to villages. The introduction fell a bit short on this. Also, I would like to see more context about the historic patterns of plant use and traditions by the Chin ethnic group. We are talking about communities whose main source of ‘traditional’ food is corn [line 432], which is a species of Central American origin, and results showed that a significant percentage of the most frequently used species are non-native to Myanmar. Therefore, four or five centuries ago, they must have used different species. If the preference for the colours (e.g., red, brown) stayed the same over time (e.g., through millennia), then which species they most used before the introduction of the species of American origin, many of which ranked in the top positions? How preserved are then those traditions? It is always difficult to talk about ‘traditions’, but more context in the introduction could help to understand the time scale of those.

Methods

It is really impressive the high number of interviews made for this study. Congratulations! Please specify the herbarium name (not just the institution) and its acronym based on Index Herbariorum. Also, were the collections made under a collecting permit? If so, please provide the information. If no collecting permit was required, please say so.

Results

It would be very useful to map the collecting sites (not just provide the geographic coordinates). I did not see the specific collection sites mapped in figure 1, only the areas of the communities. They could be added to Fig. 1.

I was expecting to find the raw information (obviously without disclosing personal information) of the 1557 surveys, to be able to replicate the results. Without this information, I have no means to verify if the indexes were correctly calculated or not, etc. One of the principles of science is repeatability. By not proving the raw data, results cannot be repeated nor verified. Also, many authors now share the R codes for anyone to replicate the analyses.

Lastly, the title of section 3 missed the comparison between “genders”, which was included in one of the research questions.

Discussion

Several species reported in the top positions of frequency of use are not native: Ageratina adenophora, Bixa orellana, and Chromolaena odorata are from South America; Curcuma longa from India; Mentha spicata from Europe; Morus nigra from Iran; Parkia javanica form Java; Hibiscus sabdariffa from Central Africa; etc. If they are not native, but commonly used, how do those fit within the traditions kept by the indigenous communities along the generations? In the case of the species of American origin, they have probably been introduced just a few hundred years ago at the most, but no information is provided about that. When was Chromolaena odorata introduced into Myanmar (or Asia)? And Bixa orellana? I think the paper could be highly improved by exploring the historic origin of those species to understand the evolution of local traditions. Only then, discussions can help to avoid neglecting to recognise that some species have only been used locally since recent times, relatively speaking.

In line 309, it says “we found that the Fabaceae and Fagaceae families had the largest number of species used for textile dyeing in the Lutuv-Chin community of Myanmar.” Then “Fabaceae and Fagaceae were the main sources of plants used for textile dyeing. The similarity may be due to the high similarity of anthropogenic landscapes, customary approach, sociocultural context, and vegetation types. However, further studies are needed to validate these hypotheses.” It is important to distinguish here between the richest families in terms of dyeing species and the most frequently used species, which could belong to not-so-species-rich families. To me, the sense of ‘importance’ is most associated with the frequency of use and quantity of product obtained, rather than the number of species within a family. However, I find that those variables (i.e., species richness and frequency of use) are usually mixed in the text. Also, I did not see an assessment of product yield (which may differ from the frequency of use). That could explain why, for instance, bark and leaves are more frequently used in certain cases.

Regarding the conservation aspects, to me, it is not important to highlight that there are 11 species of Least Concern, which means they are not under any threat status at the present. It is more important to highlight that one was registered as Data Deficient, and 18 species still need to be categorized. Therefore, the paper cannot confirm that documenting the conservation status will provide additional arguments for biodiversity protection. If from the reported species various are introduced, and the others are of Least Concern, then there is no need of investing resources to protect them. Conservation efforts should focus on the threatened species, rather than on those that fall under the least concern category.

Author Response

Cover Letter

To,

Prof. Dr. Michael Wink

Editor-in-Chief

Diversity

Subject: Submission of a revised paper entitled “Traditional knowledge of textile dyeing plants: A case study in the Chin ethnic group of western Myanmar” [Manuscript ID: diversity-1963105]

Dear Prof. Wink,

Thank you for your email dated 15-Oct-2022 enclosing the reviewers’ comments. We thank you; Section Managing Editor Mr. Mario Pei; Assistant Editor Dr. Milica Velimirovic; and the two reviewers for evaluating our manuscript entitled “Traditional knowledge of textile dyeing plants: A case study in the Chin ethnic group of western Myanmar” and providing us with constructive criticism to improve the manuscript. We have thoroughly revised all the comments and addressed all the minor corrections and the major amendments suggested by the reviewers. Detailed responses to editors’ and reviewers’ comments are given a point-by-point manner below (blue). Line numbers refer to the manuscript file. We also highlighted the corrections in Appendix 1 with yellow fill color.

We hope that the revised version is now suitable for publication and look forward to hearing from you in due course.

Sincerely,

Tial C. Ling

Reviewer 2 Report

Dear authors

the paper is related to an interesting topic and I found it very informative, by the way, there were some language corrections. I have made comments. I hope the erticle will be improved and published.

Author Response

(The authors gave the same response as above.)

Reviewer 3 Report

Very interesting research on an interesting topic - the use of natural plant dyes. Or knowledge about the use of natural plant dyes.

However, I see 2 shortcoming in the research that need addressing.

1. It makes little sense to use quantitative ethnobotanical indices when you are studying just one use - the use of a plant as a dye. Or maybe two uses: the use of a plant for extraction of dye and the use of a plant as mordant. Other than that I don't see which use categories you would have used for your calculations. Or else, you need to explain this in the paper and also include all those use categories clearly in the data table in annex. Then we understand what you consider as use categories for those calculations. If you look at the Prance et al paper you reference (ref 45), you will find that there the use categories considered are clearly defined in the data table, e.g. major & minor food, major & minor construction, technology, remedy, etc. If you only use one use category (dye), then you just count the number of times a species is listed. No need to count indices. 

It doesn't mean your research / results are not good, it just means that you don't need to calculate indices that make no sense. The quantification you can do, is indicate how many times each species has been mentioned across your informants. 

These kind of indices are typically used when a wide range of plant uses is studied, to be able to indicate which species are most 'valuable' to people (culturally salient), based on the multitude of uses. Even then, don't calculate 4 indices that indicate more or less the same. Decide which one makes sense, and shows what matters. 

2. I understand from the paper that whilst you carried out the survey in 14 communities (8 towns, 6 villages), no useful data were collected in the 8 towns (people didn't know any species used for plant dyes). So in fact, the research data only result from the surveys in 6 communities? I don't see anywhere in the paper how many of the 1557 people surveyed actually did have knowledge about plant dyes. That's important to mention since you focus the paper on the importance of preserving traditional knowledge.

A final suggestion for the paper is to indicate clearly whether you only recorded 'knowledge' about the use of natural dyes, or whether people you interviewed still use these plants to dye fabric. That is a very important distinction for traditional plant uses.

Author Response

(The authors gave the same response as above.)

Round 2

Reviewer 1 Report

The manuscript has improved considerably and I am pleased with the changes. Some minor edits are still required.  

Line 48: the verb refers to "The first written record", which is a singular noun, and therefore it should be "was" and not "were". Otherwise, change "record" to "records".

Line 77: move "(TCL personal observation)" to the end of the sentence.

There might be other small edits required. 

Thank you for reviewing the data and calculations, and for adding the data as supplementary information.

Author Response

Cover Letter

To,

Prof. Dr. Michael Wink

Editor-in-Chief

Diversity

Subject: Submission of a revised paper entitled “Traditional knowledge of textile dyeing plants: A case study in the Chin ethnic group of western Myanmar” [Manuscript ID: diversity-1963105]

Dear Prof. Wink,

Thank you for your email dated 1-November-2022 enclosing the reviewers’ comments. We thank you; Managing/Assistant Editors and the reviewers for evaluating our manuscript entitled “Traditional knowledge of textile dyeing plants: A case study in the Chin ethnic group of western Myanmar” and providing us with constructive criticism to improve the manuscript. We have thoroughly revised all the comments and addressed all the minor corrections suggested by the reviewers. Detailed responses to editors’ and reviewers’ comments are given a point-by-point manner below (blue). Line numbers refer to the manuscript file.

We hope that the revised version is now suitable for publication and look forward to hearing from you in due course.

Sincerely,

Tial C. Ling

Detailed Responses

Reviewer 1: Comments and Suggestions for Authors

The manuscript has improved considerably, and I am pleased with the changes. Some minor edits are still required.

We are thankful for the reviewer’s constructive comments and valuable suggestions that improved the manuscript. We have thoroughly and carefully read the entire documents and corrected what are supposed to be corrected.

Introduction

Line 48: the verb refers to “The first written record”, which is a singular noun, and therefore it should be “was” and not “were”. Otherwise, change “record” to “records”.

We have replaced “were” with “was”. Please see line 48.

Line 77: move “(TCL personal observation)” to the end of the sentence.

We have moved (line 78).

There might be other small edits required. Thank you for reviewing the data and calculations, and for adding the data as a supplementary information.

We truly appreciated for suggestions us to share the data in the paper. We have added some minor corrections in the new version of the manuscript. For examples, we replaced “The primary motivation the translation” with “The primary motivation was the translation”. Please find line 97-98. We used “leaves and bark” instead of “leaves, and bark” (line 134). We replaced “Warm Temperate Rainforest” with “warm temperate rainforest” (line 148-149); “We provided We excluded the reports by informants from the towns and the villagers…” with “We excluded the reports by informants from the towns and the villages…” (line 233-234); “60-85” with “60 to 85”; “are” with “were” (line 442); “Myrica esculentaBuch.Ham.” with “Myrica esculenta Buch. Ham.” (line 446-447); “deu” with “deu” (line 468); and “corn  food” with “corn food” (line 478);

Regards,

Authors

Reviewer 3 Report

It is now clear from the combined data table that the 'uses' that have been used to calculate Use Reports and Use Values are 'mordant' as well as the separate colours 'black, blue, brown, green, purple, red, yellow'. This is however still not described / indicated in the text itself. It is important this is done, else the reader will not understand what these 'uses' are. So please, in the Materials & Methods section where the calculation method of these indices is explained, insert a sentence as follows "The use categories used in this study are mordant, black, blue, brown, green, purple, red and yellow.

I also recommend that the data table 'Combined_data' is included as supplementary file with the paper. This is not yet listed in under Supplementary Materials.

Author Response

Cover Letter

To,

Prof. Dr. Michael Wink

Editor-in-Chief

Diversity

Subject: Submission of a revised paper entitled “Traditional knowledge of textile dyeing plants: A case study in the Chin ethnic group of western Myanmar” [Manuscript ID: diversity-1963105]

Dear Prof. Wink,

Thank you for your email dated 1-November-2022 enclosing the reviewers’ comments. We thank you; Managing/Assistant Editors and the reviewers for evaluating our manuscript entitled “Traditional knowledge of textile dyeing plants: A case study in the Chin ethnic group of western Myanmar” and providing us with constructive criticism to improve the manuscript. We have thoroughly revised all the comments and addressed all the minor corrections suggested by the reviewers. Detailed responses to editors’ and reviewers’ comments are given a point-by-point manner below (blue). Line numbers refer to the manuscript file.

We hope that the revised version is now suitable for publication and look forward to hearing from you in due course.

Sincerely,

Tial C. Ling

Reviewer 2: Comments and Suggestions for Authors

It is now clear from the combined data table that the ‘uses’ that have been used to calculate Use Reports and Use Values are ‘mordant’ as well as the separate colours ‘black, blue, brown, green, purple, red, yellow’. This is however still not described/indicated in the text itself. It is important this is done; else the reader will not understand what these ‘uses’ are. So please, in the Materials & Methods section where the calculation method of these indices is explained, insert a sentence as follows “The use categories used in this study are mordant, black, blue, brown, green, purple, red, and yellow.

We thank the reviewer for the precious comments and suggestions. We have taken the reviewer’s comments into account and added “The use categories used in this study are mordant, black, blue, brown, green, purple, red, and yellow” in Materials and methods (line 198-199).

I also recommend that the data table ‘Combined_data’ is included as supplementary file with the paper. This is not yet listed in under Supplementary Materials.

We have added “Rcode for indices S3” and “Combined_data_S4” as Supplementary Materials following the reviewer’s recommendation in the new version. Please see line 512-514.

Regards,

Authors